# Coherent optomechanical state transfer between disparate mechanical resonators

Matthew J. Weaver [1], Frank Buters[2], Fernando Luna[1], Hedwig Eerkens[2], Kier Heeck[2], Sven de Man[2] & Dirk Bouwmeester[1,2]

Systems of coupled mechanical resonators are useful for quantum information processing and fundamental tests of physics. Direct coupling is only possible with resonators of very similar frequency, but by using an intermediary optical mode, non-degenerate modes can interact and be independently controlled in a single optical cavity. Here we demonstrate coherent optomechanical state swapping between two spatially and frequency separated resonators with a mass ratio of 4. We find that, by using two laser beams far detuned from an optical cavity resonance, efficient state transfer is possible. Although the demonstration is classical, the same technique can be used to generate entanglement between oscillators in the quantum regime.

[1] Department of Physics, University of California, Santa Barbara, CA 93106, USA. [2] Huygens-Kamerlingh Onnes Laboratorium, Universiteit Leiden, 2333 Leiden, CA, The Netherlands. Correspondence and requests for materials should be addressed to M.J.W. (email: mweaver@physics.ucsb.edu)

Hybrid quantum systems have been developed with various mechanical, optical, and microwave harmonic oscillators[1–6]. The coupling produces a rich library of interactions including two-mode squeezing[7–10], swapping interactions[1, 3, 11, 12], back-action evasion[13, 14], and thermal control[15–17]. In a multimode mechanical system, coupling resonators of different scales (both in frequency and mass) leverages the advantages of each resonance. For example: a high frequency, easily manipulated resonator could be entangled with a low frequency massive object for tests of gravitational decoherence[18–20]. Through a process similar to STIRAP (stimulated Raman adiabatic passage)[21] in atomic physics it is possible to couple two very different mechanical resonators with an effective beam splitter interaction. We investigate this interaction, and demonstrate efficient and coherent state transfer between two frequency separated mechanical resonators in the same cavity.

Efforts are under way to control systems with several mechanical modes at the quantum level[2, 22, 23]. Hybridization and coherent swapping have been observed in optomechanical[12, 23, 24] and electromechanical[25–27] systems with nearly degenerate modes. Because the interaction between two coupled resonators decreases dramatically with frequency separation, either precise fabrication or frequency tuning is required to ensure degenerate mechanical modes. In many of these systems a separate optical cavity is necessary to control the motion of each mechanical resonator, which leads to complicated systems[12, 27]. Dynamically coupling non-degenerate resonances together in a single cavity avoids these technical difficulties, while still allowing for individual control of each resonance. In an optomechanical system where mechanical resonances are spaced further apart than the optical cavity linewidth, each resonance can be addressed independently with a laser detuned to that mechanical resonance frequency.

Here we investigate the real time dynamics of a coupled mode system and show coherent optomechanical state swapping between two mechanical modes. High swapping efficiency is possible in a region with large beam detuning from the cavity resonance. We discuss implementation of this method in the quantum regime and some capabilities of interacting quantum systems with large frequency separation.

## Results

**Optomechanical system.** Our optomechanical system consists of a room temperature Fabry–Pérot cavity with one fixed end mirror, one moving end mirror on a trampoline (resonator 1) and one trampoline membrane (resonator 2)[28–30] inside the cavity as shown in Fig. 1. The radiation pressure force on the resonators from photons in the cavity and the position dependent cavity phase shift mediate an interaction between the two resonators and the optical cavity[5]. The resonator frequencies are $\omega_1/2\pi = 297$ kHz for the end mirror and $\omega_2/2\pi = 659$ kHz for the membrane and the optical decay rate of the cavity is $\kappa/2\pi = 200$ kHz, so the system is in the resolved sideband regime.

**Optomechanical swapping.** We couple the two non-degenerate modes by modulating the inter-resonator coupling coefficient between resonators 1 and 2 at their difference frequency. Buchmann and Stamper-Kurn[31] found that an equivalent effect is produced by injecting two laser beams separated by the mechanical difference frequency into an optomechanical cavity. In the microwave regime it has been shown that driving with two tones leads to an avoided crossing of the mechanical energy levels of two resonators with different frequencies[22, 32]. Here, a single laser beam detuned from cavity resonance by the mechanical

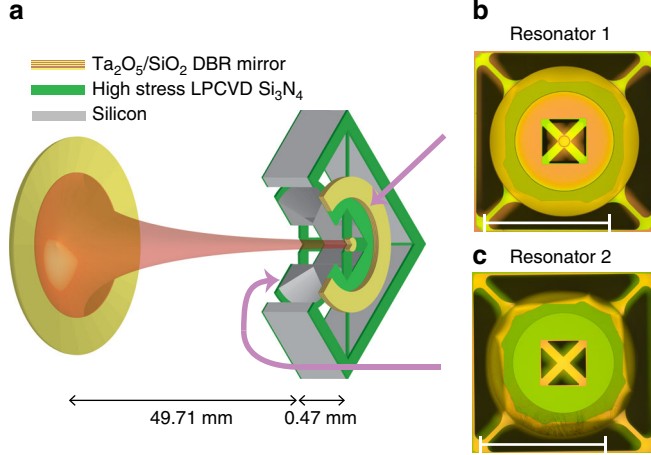

**Fig. 1** Optomechanical set-up with two resonators in an optical cavity. **a** A schematic diagram of the optical cavity with two mechanical trampoline resonators. The resonators are constructed from low pressure chemical vapor deposition (LPCVD) silicon nitride. One resonator has a distributed Bragg reflector (DBR) mirror **b** and one resonator is a bare membrane **c**. **b** and **c** are optical microscope images of the two resonators, with 1 mm scale bar. The resonators are suspended from a shared outer resonator to provide mechanical isolation from the environment. This figure is not to scale

frequency of one resonator swaps excitations between that resonator mode and the optical cavity mode[33]. A second laser beam detuned by the other mechanical frequency will concurrently swap excitations of the other resonator with the optical mode, resulting in a net swapping between the two mechanical modes. A schematic diagram of the exchange operation and the effective Λ-type system produced is shown in Fig. 2. This interaction can be described by the beam splitter Hamiltonian[31]:

$$H_{\text{int}} = \frac{J}{2}\left(b_1^\dagger b_2 + b_2^\dagger b_1\right), \qquad (1)$$

where $J$ is the optomechanical swapping rate, and $b_j$ is the annihilation operator for the $j$th mechanical mode.

To investigate this interaction, we prepare one resonator in an excited state and then observe the swapping dynamics of the coupled system. We excite resonator 2 into a large coherent state by applying a voltage at its resonance frequency to an electrode behind the sample and then turn on the two laser beams. Figure 3 shows the measured amplitude of motion of the two resonators. We observe in real time as the mechanical excitation is swapped back and forth between the two resonators in a repeatable fashion. Figure 3b shows the response to a single optical swapping interaction. The operation can be modeled as an underdamped exchange between two coupled harmonic oscillators, and the fits indicate that our system operates in this regime (Methods section). The motion dips down to the thermal fluctuation level every time the state is exchanged, indicating complete state swapping. We now investigate the efficiency of the system and its coupling to different loss baths.

**Power and detuning dependence of swapping parameters.** If the transfer rate, $J$, is much slower than the mechanical frequencies, the classical amplitudes of the modes $b_1$ and $b_2$ evolve slowly. Under this approximation the transfer rate, $J$, and total loss rate $\Gamma$

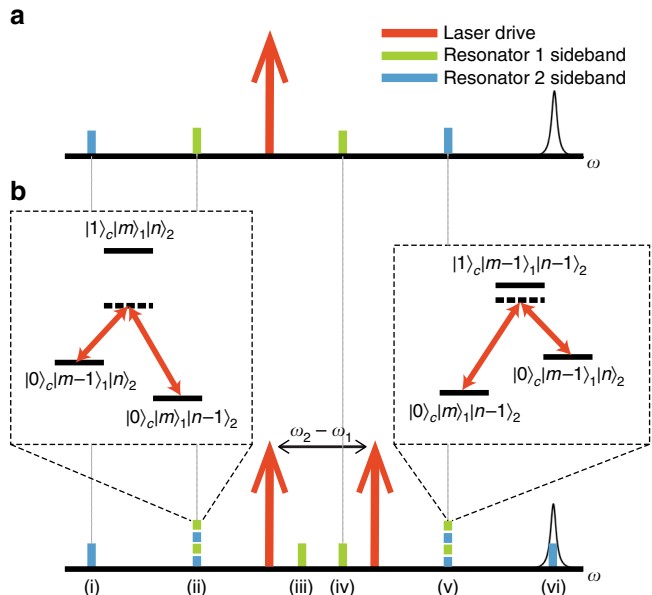

**Fig. 2** Generating coupling between two resonators with two laser drives. **a** A single laser drive (red arrow) sent into the cavity produces four sidebands, two for each resonator. The laser is detuned from a cavity resonance on the right. **b** A second laser can be added to generate optical swapping. (ii) and (v) are overlapping sidebands of the two resonators. The insets indicate the analogy to state transfer in an atomic Λ-type system. The quantum number states are the photon occupation of the cavity, phonon occupation of resonator 1 and phonon occupation of resonator 2. Detuning from the intermediary state avoids losses due to light leaking out of the cavity. (iii) and (iv) are the unmatched sidebands of resonator 1 and (i) and (vi) are the unmatched sidebands of resonator 2. By adjusting the laser detuning, the sidebands (i–vi) can be separately aligned with the cavity resonance to interact with one resonator at a time or both at once. In the case shown here, the state of resonator 2 is swapped with the cavity, because sideband (vi) is aligned to the cavity. This figure is not to scale

are given by:

$$J = 2g_1g_2\sqrt{n_1n_2}\left(\frac{\overline{\omega}-\overline{\Delta}}{\kappa^2/4+\left(\overline{\omega}-\overline{\Delta}\right)^2}-\frac{\overline{\omega}+\overline{\Delta}}{\kappa^2/4+\left(\overline{\omega}+\overline{\Delta}\right)^2}\right), \quad (2)$$

$$\Gamma = \sum_{i,j=1,2}\frac{n_ig_j^2\kappa}{\kappa^2/4+\left(\Delta_i-\omega_j\right)^2}-\frac{n_ig_j^2\kappa}{\kappa^2/4+\left(\Delta_i+\omega_j\right)^2}+\frac{\gamma_j}{2}, \quad (3)$$

$$n_i = \frac{P_{\rm in}}{2\hbar\omega_{\rm Li}}\frac{\kappa_{\rm ex}}{\kappa^2/4+\Delta_i^2}, \quad (4)$$

where $g_j$, $\omega_j$, and $\gamma_j$ are the single photon optomechanical coupling rate, mechanical frequency and mechanical damping rate of the $j$th mode. $\Delta_i$ and $n_i$ are the detuning to the red side and cavity photon number of the $i$th cavity mode. $\overline{\Delta}$ and $\overline{\omega}$ are the mean detuning and mean frequency of the two modes. $\omega_{\rm Li}$ is the laser frequency of the $i$th beam, $\kappa_{\rm ex}$ is the input coupling rate and $P_{\rm in}$ is the input optical power. The swapping rate, $J$, is the sum of two Fano-like resonances from each set of matched sidebands. These exchange the mechanical state through a virtual state near the optical cavity resonance as pictured in the two insets in Fig. 2b. The Lorentzian resonances in the expression of the loss rate, $\Gamma$, are the optically induced loss or gain of the $j$th mode due to the $i$th laser beam. There is one term for each of the eight sidebands (Fig. 2a, b). The complete model is given in Methods section.

Both optomechanical gain and loss should be avoided, as gain can introduce noise into the system. Because $\Gamma$ decreases more quickly than $J$ with increasing $\overline{\Delta}$, the ideal detuning is on the red side of the cavity, far from all resonances, in a region with negligible optomechanical amplification. Figure 4 shows an exploration of state swapping in a region with large detuning. The range is limited to regions of coherent swapping, where $J > \Gamma$. We observe the expected dependencies on detuning and input power for the coupling and loss rates. For smaller detunings the dominant loss is residual optical cooling of resonator 2, a byproduct of its unmatched red sideband. For large detunings mechanical leakage to the environment dominates, and the peak efficiency is in the middle at $\overline{\Delta}/2\pi = 2.3$ MHz.

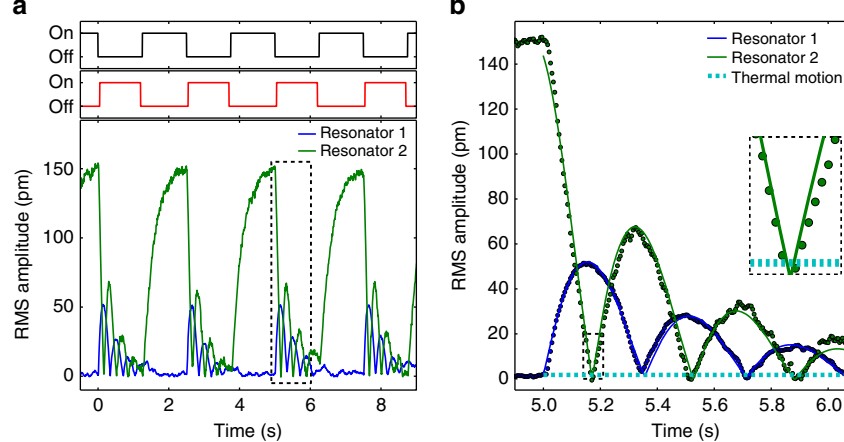

**Fig. 3** Optomechanical swapping between mechanical resonators. **a** We alternate turning on a mechanical drive (black) and an optical swapping field (red), while continuously measuring the root mean square (RMS) amplitude of motion of the two resonators. This single-shot measurement shows the repeatable dynamics of the system. **b** A single swapping interaction (the dashed box in **a**) shows phonon Rabi oscillations. Solid lines are fits to the measured data points, and the dotted line indicates the thermal motion of the two resonators. Because the motion dips down to the thermal noise level every period, there is complete state swapping. The inset shows one such dip after a single complete state transfer

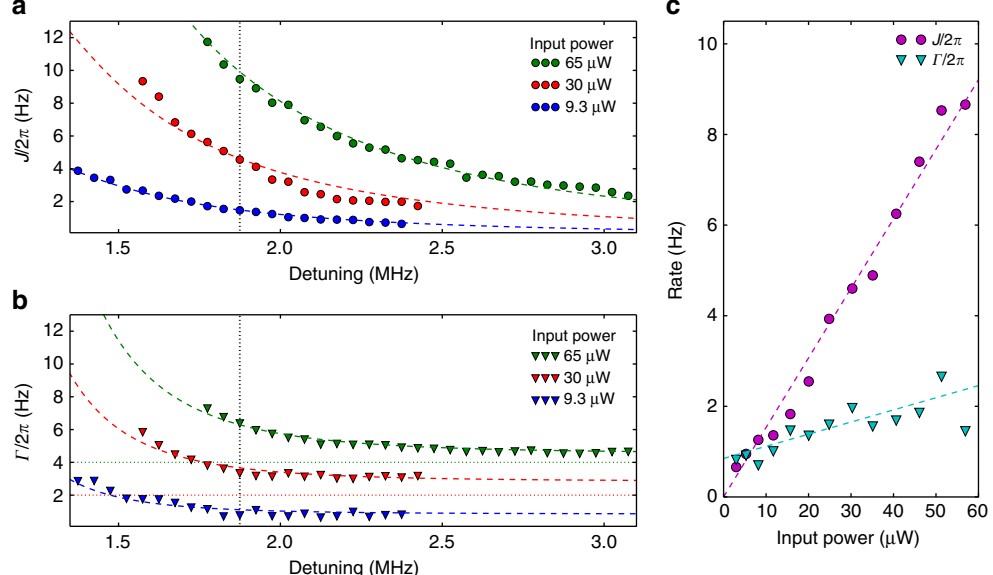

**Fig. 4** Parameter dependence of optomechanical swapping rate and total loss rate. Optomechanical swapping rate, $J$ **a**, and total loss rate, $\Gamma$ **b**, are measured as a function of detuning ($\bar{\Delta}/2\pi$). The dashed lines are two parameter fits based on Equations (2) and (3). For clarity the higher power measurements of $\Gamma$ are vertically offset by 2 and 4 Hz as indicated by the dotted lines. **c** $J$ and $\Gamma$ are measured as a function of input power at a detuning of 1.87 MHz (indicated by black dotted line in **a** and **b**). The dashed lines are two parameter fits based on Equations (2) and (3). The ratio between the measured optical power and the input power $P_{in}$ and the mean bare mechanical dissipation rate $(\gamma_1 + \gamma_2)/2$ are the fitting parameters (Methods section). Statistical uncertainties are smaller than the point size

## Discussion

Two useful operations in a quantum network of oscillators are a complete state transfer ($\pi$-pulse) and a partial state transfer ($\pi/2$-pulse) to generate an entangled state. If we terminate the swapping after one of these pulses, 58% of the phonon occupation is conserved in a $\pi$-pulse and 77% of the occupation is conserved in a $\pi/2$-pulse (Methods section). The swapping rate demonstrated here at room temperature is not sufficient to overcome the large thermal decoherence rate ($n_{th}\ \gamma$) from the environment even at millikelvin temperatures. However, both the efficiency of transfer and the swapping rate could be improved significantly by decreasing the cavity loss. The finesse of our cavity is currently limited by absorption in the membrane trampoline, and we estimate that using a thinner membrane would improve the finesse by at least a factor of four. Most of the detunings close to the cavity resonance are in the overdamped regime, where energy transfer is only possible with large losses. With an increased finesse, a point close to the cavity resonance appears where the positive and negative components of $\Gamma$ cancel, leading to nearly lossless classical state transfer (>99% efficiency). In the quantum regime, the negative component of $\Gamma$ introduces extra decoherence, so the quantum state transfer is more limited (56% efficiency). However, the effects of coherent swapping should still be visible (Methods section.)

Although, we have focused on swapping states between the fundamental modes of two resonators, the technique is general and can also be applied to higher order modes of the same resonator. We apply the exact same scheme to swap energy between the fundamental ($\omega_1/2\pi = 659$ kHz) and the first excited ($\omega_2/2\pi = 1199$ kHz) mode of the membrane trampoline (data in Supplementary Fig. 1). Sequential swapping pulses between many mechanical modes in a cavity could generate a large network of coupled modes. Each mode is individually addressable because of its frequency separation from the other modes. Low frequency resonators with long mechanical lifetimes could serve as storage for quantum information generated with a high frequency resonator.

This technique can also be used to study quantum mechanics in a high-mass system. Larger systems tend to suffer from small optomechanical coupling rates and slow interactions. We can instead prepare a quantum superposition state in a high frequency resonator with large optomechanical coupling and transfer it into the high-mass resonator. After letting the system evolve for an extended period, then transferring the motion back to the high frequency resonator, we can determine if the state decohered. Finally, this work could be extended to provide directional adiabatic transfer of states with STIRAP by using separate time-varying intensity pulses for the two input laser beams[21].

In conclusion, exchange of mechanical energy between modes which are naturally uncoupled opens up many possibilities in quantum and classical physics. We have investigated the real time dynamics of such a system. We demonstrate that despite the many loss effects present, efficient coherent state transfer between two spatially and frequency separated mechanical resonators is possible. These results can be extended to the quantum regime to investigate quantum effects with many diverse mechanical oscillators.

The authors would like to acknowledge a related manuscript which appeared during the completion of this manuscript[34].

## Methods

**Optomechanical system**. The optomechanical system is an extension of previous systems[35]. We use a Fabry–Pérot cavity with one fixed end mirror with a nominal radius of curvature of 50 mm. The other side of the cavity is formed by two trampolines fabricated on opposite sides of a tethered silicon block (Fig. 1). The block acts as a mechanical low pass filter and provides greater than 65 dB of vibration isolation from the environment[36]. The cavity alignment uses the same technique used for single trampoline resonators[35]. Four piezo motors adjust the cavity in-coupling and three motors align the cavity itself. The DBR mirror on the trampoline is only 75 μm in diameter, so we align the beam waist of the cavity mode close to the DBR to avoid clipping losses. Mode calculations indicate that the beam radius should be ~ 16 μm at both the DBR and bare membrane trampoline. Based on the free spectral range of the cavity, we estimate an exact length of 50.18 mm. The cavity is slightly longer than 50 mm because the mismatch in stress between the silicon nitride and the DBR mirror leads to a slight inward curvature

with a radius of ~1.5 mm[37]. Because the two resonators are fabricated on the same chip, no extra alignment is needed for the additional membrane trampoline in the middle. This technique could be extended to even more resonators by attaching multiple chips together.

The system behaves as the sum of its two constituent parts: a traditional optomechanical cavity with a single moving end mirror and a membrane in the middle system[30]. A membrane in the middle system has a finesse which depends on the position of the membrane with respect to the nodes of the cavity[38, 39]. Supplementary Fig. 2a shows a periodic finesse response as, we vary the node position by changing wavelength. The optical cavity loss is dominated by absorption in the membrane trampoline. We numerically model the system with the transfer matrix method[40] and extract the imaginary refractive index ($n_{im} = 3.2 \times 10^{-5}$) of the $Si_3N_4$ membrane and the chip thickness (470 μm.) Both values match expectations[38]. The nitride we use is about 10 times thicker than many other membrane in the middle set-ups[4, 28–30], so we can likely reduce optical losses with a thinner membrane. We have achieved finesses up to 180,000 in the same set-up without the membrane present[36].

We also investigate the optomechanics of each individual mode. Supplementary Fig. 2b shows the optical damping of each resonator as a function of detuning. The damping can be modeled perfectly using the linear optomechanical Hamiltonian for a single resonator[5], indicating that with a single laser beam the modes can be treated independently. From these measurements and others, we extract the optical decay rate, $\kappa/2\pi = 200 \pm 10$ kHz, the mechanical frequencies $\omega_1/2\pi = 297$ kHz and $\omega_2/2\pi = 659$ kHz, the mechanical damping rates $\gamma_1/2\pi = 1.5 \pm 0.1$ Hz and $\gamma_2/2\pi = 1.0 \pm 0.1$ Hz, and the single photon optomechanical coupling rates $g_1/2\pi = 0.9 \pm 0.1$ Hz and $g_2/2\pi = 1.3 \pm 0.1$ Hz. From finite element analysis simulations, we determine that the effective masses are $\sim m_1 = 150$ ng and $m_2 = 40$ ng.

**Fabrication.** The fabrication process is a slight modification of the procedure for nested trampoline resonators[36]. We summarize here: 450 nm of LPCVD (low pressure chemical vapor deposition) high stress silicon nitride is deposited on both sides of a silicon wafer, followed by a commercial $SiO_2/Ta_2O_5$ DBR mirror on the front and a $SiO_2/SiN$ layer on the back. The mirror is etched with inductively coupled plasma (ICP) $CHF_3$ into disks for the cavity end mirror and a protective ring. The back $SiO_2/SiN$ films are etched with $CHF_3$ ICP into a protective ring. The silicon nitride layers on both sides are then etched with $CF_4$ to produce the front and back side trampolines. The silicon underneath the devices is removed with a deep reactive ion etch, followed by an etch in TMAH (tetramethylammonium hydroxide) solution. The devices are dipped in buffered HF to remove the top protective layer of $SiO_2$ from the mirror.

**Experimental procedure.** We now turn to the generation of optomechanical state swapping. We use a two laser scheme as depicted in Supplementary Fig. 3. One laser is locked to the cavity resonance with the Pound–Drever–Hall technique[41] using an avalanche photo diode as a detector, and the error signal is sent to two lock-in amplifiers, each of which monitors one mechanical frequency and extracts the amplitude of motion of the corresponding resonator. Before the swapping experiment shown in Fig. 3 is performed, we calibrate the mechanical motion of the devices by measuring the thermal motion for ~1 min. The optomechanical gain rate is less than 20% of the mechanical damping rate, and hence we do not expect or observe notable contributions to the noise from optomechanics. Another laser is passed through an acousto-optic modulator (AOM) with an RF drive that we modulate fully at half the mechanical difference frequency. The first order diffracted mode contains the two frequencies that we use to drive optomechanical swapping in the cavity. We have verified that the carrier frequency is completely suppressed and that higher harmonics are insignificant with cavity transmission measurements. We cannot measure the optical input power directly, so we split off some power before the cavity to measure. Finally, a ring electrode behind the outer resonator is used to excite the motion of the trampoline resonators using the dielectric force from the gradient of the electric field[42].

We repeat this experiment for many powers and detunings, and extract the swapping rate and loss rate for each instance. The unmatched sidebands in Fig. 2b produce loss, but they also shift the frequencies of the two mechanical resonances. Therefore, when performing the detuning and power sweeps shown in Fig. 4 the spacing between the two laser beams must be continuously adjusted to match the mechanical difference frequency. The readout laser can optomechanically decrease or increase the bare mechanical linewidths of the resonators a small amount depending on the lock settings. We therefore fit the mean bare mechanical linewidth and the ratio between the measured optical power and the input power for every sweep shown in Fig. 4. We also perform a swapping experiment using the two lowest order modes of the membrane trampoline to verify that the exact same scheme works for a single membrane in the middle. The swapping is shown in Supplementary Fig. 1.

**Two-tone swapping interaction theory.** Because the experiment performed here is entirely classical, we limit ourselves to the classical optomechanical equations of motion following a similar path to Shkarin et al.[23]. However, the results can be generalized to the quantum regime[31]. The linearized equations of motion for the

cavity field fluctuations, $a$, and mechanical displacements, $b_1$ and $b_2$, are given by:

$$\dot{a} = -\left(\frac{\kappa}{2} + i\omega_c\right)a + \sum_j i\frac{g_j a}{x_{zpm}}\left(b_j + b_{j^*}\right) \tag{5}$$

$$+ \sqrt{\kappa_{ex}}\left(a_{in1}e^{-i(\omega_c + \Delta_1)t} + a_{in2}e^{-i(\omega_c + \Delta_2)t}\right),$$

$$\dot{b}_j = -\left(\frac{\gamma_j}{2} + i\omega_j\right)b_j + ig_j a^* a. \tag{6}$$

After some algebraic manipulation, we arrive at the following equations for the adiabatic time evolution of the amplitude of the two resonators:

$$\dot{b}_1 = \left(-\frac{\gamma_{1tot}}{2} + i\delta\omega_1\right)b_1 + \left(-\frac{\gamma_{12}}{2} + i\frac{\bar{J}}{2}\right)b_2, \tag{7}$$

$$\dot{b}_2 = \left(-\frac{\gamma_{2tot}}{2} + i\delta\omega_2\right)b_2 + \left(-\frac{\gamma_{12}}{2} + i\frac{\bar{J}}{2}\right)b_1, \tag{8}$$

$$\gamma_{jtot} = \gamma_j + \sum_{i=1,2}\frac{2n_i g_j^2 \kappa}{\kappa^2/4 + (\Delta_i - \omega_j)^2} - \frac{2n_i g_j^2 \kappa}{\kappa^2/4 + (\Delta_i + \omega_j)^2}, \tag{9}$$

$$\delta\omega_j = \sum_{i=1,2}\frac{n_i g_j^2(\Delta_i - \omega_j)}{\kappa^2/4 + (\Delta_i - \omega_j)^2} - \frac{n_i g_j^2(\Delta_i + \omega_j)}{\kappa^2/4 + (\Delta_i + \omega_j)^2}, \tag{10}$$

$$\tilde{J} = 2g_1 g_2 \sqrt{n_1 n_2}\left(\frac{\overline{\omega} - \overline{\Delta}}{\kappa^2/4 + (\overline{\omega} - \overline{\Delta})^2} - \frac{\overline{\omega} + \overline{\Delta}}{\kappa^2/4 + (\overline{\omega} + \overline{\Delta})^2}\right), \tag{11}$$

$$\gamma_{12} = g_1 g_2 \sqrt{n_1 n_2}\left(\frac{\kappa}{\kappa^2/4 + (\overline{\omega} - \overline{\Delta})^2} - \frac{\kappa}{\kappa^2/4 + (\overline{\omega} + \overline{\Delta})^2}\right). \tag{12}$$

Although these equations look complex, they can be matched up term for term with the effects of each sideband. $\gamma_{jtot}$ and $\delta\omega_j$ are the optical damping and optically induced frequency shift on the $j$th resonator due to the $i$th beam in the cavity. There are eight of these terms total, one for both sidebands on both lasers from both resonators. $\bar{J}$ and $\gamma_{12}$ are the bare optomechanical transfer rate and the loss induced decrease in the transfer rate. The first term in $\bar{J}$ is produced as the net effect of two optomechanical swapping interactions with the cavity as depicted in the right inset of Fig. 2b. The second term in $\bar{J}$ is produced by two optomechanical two-mode squeezing interactions with the cavity (left inset of Fig. 2b). If we absorb the frequency shifts into $b_1$ and $b_2$, the solutions are of the following form:

$$b_1(t) = c_1 e^{-\Gamma t/2}\left|\sin\left(\frac{Jt}{2}\right)\right|, \tag{13}$$

$$b_2(t) = c_2 e^{-\Gamma t/2}\left|\cos\left(\frac{Jt}{2}\right)\right|, \tag{14}$$

$$J = \sqrt{\bar{J}^2 - \frac{\gamma_{12}^2 + (\gamma_{1tot} - \gamma_{2tot})^2}{2}}, \tag{15}$$

$$\Gamma = \frac{\gamma_{1tot}}{2} + \frac{\gamma_{2tot}}{2}, \tag{16}$$

where $c_1$ and $c_2$ are constants dependent on the initial conditions of the system. When we apply the swapping pulses to the optical cavity, we see decaying oscillations which can be fitted precisely with the above equations. For large detunings where $J > \Gamma$, $J$ is approximately $\bar{J}$, so we treat them interchangeably in the main text.

We define the classical efficiency of an exchange pulse as the total number of phonons in the system after the pulse divided by the initial number of phonons in resonator 2. The efficiency of a $\pi$-pulse is $\exp(-\pi\,\Gamma/J)$ and the efficiency of a $\pi/2$-pulse is $\exp(-\pi\,\Gamma/2\,J)$. The efficiency of a $\pi$-pulse both theoretically and experimentally is plotted in Supplementary Fig. 4 as a function of detuning. A number of regions are inaccessible, because the optical damping is too large, and $J$ becomes imaginary. In these overdamped regions, energy can still be transferred, but there is no coherent state transfer. If the optical cavity losses are reduced by a factor of four, more regions of small detuning would become accessible.

Thus far we have focused on the losses in the system, or the positive contributions to $\Gamma$. However, $\Gamma$ has some contributions which are negative and correspond to parametric driving of the system. Parametric driving leads to an exponential increase in the motion of the resonators and is therefore equally as

unsuited to efficient state transfer as configurations with large loss. However, it is possible to find detunings for which the heating and cooling contributions cancel, and $\Gamma$ goes to zero. For these detunings classical state transfer is lossless, and the efficiency of state transfer goes to 1. In the current system such cancelation points only exist on the blue side of the cavity where the system is inherently unstable. However, if the cavity losses were reduced, a cancelation point appears on the red side, indicated by the star in Supplementary Fig. 4b. At this point the driving due to one laser beam just on the blue side of the cavity resonance is canceled by the cooling due to the other laser close to the red sideband of resonator 1. This leads to significantly higher classical efficiency ($>99\%$) and faster state transfer ($J=18$ kHz).

In the quantum regime, calculations of the efficiency are more complicated. The parametric driving, which can allow for efficient classical transport, also introduces extra noise. Furthermore, quantum states with small phonon occupation have a large thermalization rate due to the high thermal occupation of the bath, even when the resonator is cooled down to millikelvin temperatures. In Supplementary Fig. 4b, we compare the classical and quantum efficiencies. At small detunings the quantum efficiency is limited by parametric driving and at large detunings by thermalization. Supplementary Fig. 4b also assumes a bath temperature of 10 mK and an improved linewidth of 10 mHz, which is in line with the improvements seen at cryogenic temperatures for many silicon nitride devices[43, 44]. These improvements should be enough to start using this protocol in the quantum regime.

**Data availability**. All data are available from the authors upon request.

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

## Acknowledgements

The authors would like to thank W. Loeffler for helpful discussions. This work is part of the research program of the Foundation for Fundamental Research (FOM) and of the NWO VICI research program, which are both part of The Netherlands Organisation for Scientific Research (NWO). This work is also supported by the National Science Foundation Grant Number PHY-1212483.

## Author contributions

M.J.W. wrote the manuscript with input from all authors and performed the experiment with assistance from F.B., F.L. fabricated the device used in the experiment. F.B., H.E.,

S.d.M., and K.H. created the optomechanical system used for this study. D.B. supervised the entire process.

## Additional information

**Competing interests:** The authors declare no competing financial interests.

