## [Peer Review File · Nature Communications]

Reviewers' comments:

Reviewer #1 (Remarks to the Author):

the manuscript presents the results of an experiment in the field of cavity opto-mechanics, using reflective and refractive micro-oscillators in a high-finesse optical cavity. The coherent oscillation of a mechanical mode, preliminarily excited, is periodically transferred to (and recovered from) a second mechanical mode by means of opto-mechanical interaction with two properly detuned laser fields. The result reminds a bit the Rabi oscillations in an atomic system. The experiment is performed in the classical regime (i.e., just the coherent oscillation is transferred, not the quantum fluctuations, that are indeed overwhelmed by thermal noise).

The main interest of the experiment is probably to be found in its possible extension to the quantum domain. With the described scheme, applied to a system with lower thermal noise, and maybe integrated by some quantum-friendly detection technique, one could realize entangled mechanical states, quantum information transfer and storage, etc.

The experiment is scientifically sound, and its explanation is convincing. It is described very clearly, I just have few minor concerns on the manuscript (see below). I judge that the work is likely to have an impact in the community active on quantum opto-mechanics, even if I would not consider the present results as a major physical achievement (actually, they just concerns coherent dynamics in the classical regime).

On the whole, considering the aim of the journal ("Nature Communications is committed to publishing important advances of significance to specialists within each field"), I am in favor of its publication.

Minor remarks on the text.

- 1) Figure 1c shows a 50.5mm long cavity (50+0.5mm). The radius of the bulk mirror is not mentioned (it would be better to quote it), but I imagine that it is, nominally, 50mm (something between 50 and 51mm is very unlikely, and a much greater radius would give a too large waist with respect to the DBR mirror diameter that is, according to Ref.[31], below 0.1mm). However, such a cavity would be unstable. I imagine that the total cavity length is indeed a bit less than 50mm, and Fig. 1c is just inaccurate. The authors should amend it (and give more details on the cavity length, on the expected or measured beam waist on the DBR mirror, on the procedure to control the cavity length and alignment: with such a small DBR mirror, it is not straightforward).
- 2) It is not clear how the AOM generates the two beams. In Fig. S2, it is driven by an oscillator at $(\omega_2 - \omega_1)/2$. I suspect that this oscillator somehow modulates the rf driving the AOM, generating a couple of sidebands spaced by $(\omega_2 - \omega_1)$. However, I wonder if the carrier is still present (i.e., there is also a tone at the mean frequency between the two beams mentioned in the text), and if there are higher harmonics (or the modulation scheme is different....???)
- 3) Ref. [31] is lacking of the page number.

Reviewer #2 (Remarks to the Author):

Main results of manuscript

The primary result is efficient coupling of mechanical oscillators with different parameters (mass ratio ~ 4 and frequency ratio ~ 0.5). Coherent states are transferred between the mechanical oscillators with $\sim 60\%$ efficiency at a rate of ~ 10 Hz. Projections are made for reaching 99% efficiency, and increasing transfer rate to operate in the quantum regime.

Recommendation

I am concerned about the treatment of added noise and amplification. If these concerns can be adequately addressed, I am enthusiastic about recommending this manuscript for publication.

Review

While mechanical oscillators with similar operating parameters have been classically coupled, coupling of disparate oscillators is only now being investigated experimentally. The approach in this manuscript is appealing, results in high transfer efficiencies, and appears to have a reasonable outlook for reaching the quantum regime.

These results will capture significant attention from the optomechanics community. In addition, it is reasonable to expect the broader scientific community to understand the notion of coupling mechanical oscillators, and to be interested in the results — particularly insofar as there is an outlook for quantum-mechanical effects.

However, some of the discussion of the role of amplification, and its impact on efficiency, are troubling. In short, the presence of gain should not in general be counted towards efficiency, especially if one has quantum applications in mind.

The authors project 99% efficiency, but in a regime where loss is balanced out by gain. In the context of state transfer, gain is necessarily accompanied by added noise. In the context of transferring a quantum state, even a small amount of added noise may result in a mixed state. Of course — if one operates in the quantum regime — apparent added noise must be balanced by correlations in the other mechanical mode. Whether or not this is useful though, would seem to depend on the application.

To clarify the treatment of gain and added noise in this manuscript, the following issues must be addressed.

1. Are the data in Fig. 3 taken in a regime with zero amplification? If not, this must be stated clearly and the gain quantified.
2. Was the added noise during the transfer process measured? If no, both the thermal contribution and contribution due to gain should be estimated and reported.
3. In general, it's not appropriate to count gain towards efficiency if one has quantum applications in mind. Gain adds noise, which results in mixed states. Gain should be divided out of reported efficiencies.
4. Similarly, in future projections, the 99% efficiency should have gain divided out. Perhaps in the quantum regime, gain may be desirable as it implies entanglement with the other mechanical mode, but this is a separate issue from state transfer and should be discussed separately.
5. In Fig. 3b, the Resonator 1 curve first peaks at ~ 50 pm. For identical oscillators, one expects it to peak around ~ 100 pm. Is the balance explained by a difference in mechanical parameters? Naively, Resonator 1 has the lower frequency, so I expect its first peak to be higher, not lower.

Reviewer #3 (Remarks to the Author):

The authors report the realization of coupling non-degenerate mechanical oscillators using a shared optical cavity mode. In addition to the coupling of two non-degenerate fundamental modes they demonstrate coupling involving mechanical modes with higher frequency.

These results are of great importance to the optomechanics community for the first demonstration of coherently mediated coupling, but the impact extends beyond optomechanics, as set-ups modelled after these results may be useful in future high-precision measurements and in experiments investigating collapse models of quantum mechanics.

I judge the impact of their results to be sufficient to warrant publication in Nature communications, if the authors clarify some important details.

Most importantly, the authors claim their transfer to be coherent. However, the data they show are measurements of the mechanical mode's energies. Moreover, neither the main body, the methods or supplemental section contain the procedure through which these energies are obtained. This is unsatisfactory and needs to be addressed.

Related to this question, the authors mention in the main body that the thermal environment of the system lies in the mK range. How is it cooled ? Did the authors try to further cool the mechanical degrees of freedom using sideband cooling ? Their system is in a regime where this would be an efficient technique.

The thermal noise floor in Fig. 3b is not discernible, it would be helpful to have an inset around the time of one completed transfer. In the same figure, are the lines from the theory model presented ? what causes the initial offset ?

In fig. 4 data is compared to theory curves resulting from a "2 parameter fit". Which parameters were used as fitting parameters ? All parameters in the presented model can be measured independently and the authors suggest that they in fact did measure them independently.

If these comments will be taken care of, I will be happy to support publication of the author's results.

Reviewer #1 (Remarks to the Author):

the manuscript presents the results of an experiment in the field of cavity opto-mechanics, using reflective and refractive micro-oscillators in a high-finesse optical cavity. The coherent oscillation of a mechanical mode, preliminarily excited, is periodically transferred to (and recovered from) a second mechanical mode by means of opto-mechanical interaction with two properly detuned laser fields. The result reminds a bit the Rabi oscillations in an atomic system. The experiment is performed in the classical regime (i.e., just the coherent oscillation is transferred, not the quantum fluctuations, that are indeed overwhelmed by thermal noise).

The main interest of the experiment is probably to be found in its possible extension to the quantum domain. With the described scheme, applied to a system with lower thermal noise, and maybe integrated by some quantum-friendly detection technique, one could realize entangled mechanical states, quantum information transfer and storage, etc.

The experiment is scientifically sound, and its explanation is convincing. It is described very clearly, I just have few minor concerns on the manuscript (see below). I judge that the work is likely to have an impact in the community active on quantum opto-mechanics, even if I would not consider the present results as a major physical achievement (actually, they just concerns coherent dynamics in the classical regime).

On the whole, considering the aim of the journal ("Nature Communications is committed to publishing important advances of significance to specialists within each field"), I am in favor of its publication.

We thank the reviewer for the detailed comments and the positive recommendation. The results are analogous to Rabi oscillations in an atomic system and we agree that the method is likely to have many applications in the scientific community. We are also excited about extensions into the quantum regime.

Minor remarks on the text.

1) Figure 1c shows a 50.5mm long cavity (50+0.5mm). The radius of the bulk mirror is not mentioned (it would be better to quote it), but I imagine that it is, nominally, 50mm (something between 50 and 51mm is very unlikely, and a much greater radius would give a too large waist with respect to the DBR mirror diameter that is, according to Ref.[31], below 0.1mm). However, such a cavity would be unstable. I imagine that the total cavity length is indeed a bit less than 50mm, and Fig. 1c is just inaccurate. The authors should amend it (and give more details on the cavity length, on the expected or measured beam waist on the DBR mirror, on the procedure to control the cavity length and alignment: with such a small DBR mirror, it is not straightforward).

We agree with the reviewer that we did not use the exact cavity length in Figure 1c (now Figure 2a.) This has been corrected. We have also added the following text to the supplementary materials to clarify the alignment procedure and final configuration:

The cavity alignment uses the same technique used for single trampoline resonators. Four piezo motors adjust the cavity in-coupling and three motors align the cavity itself. The DBR mirror on the trampoline is only 75 μm in diameter, so we align the beam waist of the cavity mode close to the DBR to avoid clipping losses. Mode calculations indicate that the beam radius should be approximately 16 μm at both the DBR and bare membrane trampoline. Based on the free spectral range of the cavity we estimate an exact length of 50.18 mm. The cavity is slightly longer than 50 mm because the mismatch in stress between the silicon nitride and the DBR mirror leads a slight inward curvature with a radius of approximately 1.5 mm.

2) It is not clear how the AOM generates the two beams. In Fig. S2, it is driven by an oscillator at $(\omega_2 - \omega_1)/2$. I suspect that this oscillator somehow modulates the rf driving the AOM, generating a couple of sidebands spaced by $(\omega_2 - \omega_1)$. However, I wonder if the carrier is still present (i.e., there is also a tone at the mean frequency between the two beams mentioned in the text), and if there are higher harmonics (or the modulation scheme is different....???)

This is indeed the method we use to generate the two beams. We have added the following clarification to the supplementary material:

Another laser is passed through an acousto-optic modulator (AOM) with an RF drive that we modulate fully at half the mechanical difference frequency. The first order diffracted mode contains the two frequencies that we use to drive optomechanical swapping in the cavity. We have verified that the carrier frequency is completely suppressed and that higher harmonics are insignificant with cavity transmission measurements.

3) Ref. [31] is lacking of the page number.

This has been corrected.

Reviewer #2 (Remarks to the Author):

Main results of manuscript

The primary result is efficient coupling of mechanical oscillators with different parameters (mass ratio ~ 4 and frequency ratio ~ 0.5). Coherent states are transferred between the mechanical oscillators with $\sim 60\%$ efficiency at a rate of ~ 10 Hz. Projections are made for reaching 99% efficiency, and increasing transfer rate to operate in the quantum regime.

Recommendation

I am concerned about the treatment of added noise and amplification. If these concerns can be adequately addressed, I am enthusiastic about recommending this manuscript for publication.

Review

While mechanical oscillators with similar operating parameters have been classically coupled, coupling of disparate oscillators is only now being investigated experimentally. The approach in this manuscript is appealing, results in high transfer efficiencies, and appears to have a reasonable outlook for reaching the quantum regime.

These results will capture significant attention from the optomechanics community. In addition, it is reasonable to expect the broader scientific community to understand the notion of coupling mechanical oscillators, and to be interested in the results — particularly insofar as there is an outlook for quantum-mechanical effects.

However, some of the discussion of the role of amplification, and its impact on efficiency, are troubling. In short, the presence of gain should not in general be counted towards efficiency, especially if one has quantum applications in mind.

The authors project 99% efficiency, but in a regime where loss is balanced out by gain. In the context of state transfer, gain is necessarily accompanied by added noise. In the context of transferring a quantum state, even a small amount of added noise may result in a mixed state. Of course — if one operates in the quantum regime — apparent added noise must be balanced by correlations in the other mechanical mode. Whether or not this is useful though, would seem to depend on the application.

We thank the reviewer for their careful consideration of the manuscript. The distinction between the effect of gain in the quantum and classical regimes is important. By addressing this point in detail we have strengthened the manuscript and raised some interesting differences which can be studied at the classical to quantum transition. We agree that our study of coupling disparate resonators will be of interest to the scientific community, and we also find the prospect of performing the same experiment in the quantum regime exciting.

To clarify the treatment of gain and added noise in this manuscript, the following issues must be addressed.

1. Are the data in Fig. 3 taken in a regime with zero amplification? If not, this must be stated clearly and the gain quantified.

The data in Fig. 3 were taken in a regime with negligible optomechanical amplification. We have added the following clarifying remark to the main text:

Both optomechanical gain and loss should be avoided, as gain can introduce noise into the system. Because Γ decreases more quickly than J with increasing $\bar{\Delta}$, the ideal detuning is on the red side of the cavity, far from all resonances, in a region with negligible optomechanical amplification.

2. Was the added noise during the transfer process measured? If no, both the thermal contribution and contribution due to gain should be estimated and reported.

From theoretical calculations of the gain, we expect the contribution of added noise from the gain to be much smaller than the thermal contribution. We add the following statement to the supplementary material:

The optomechanical gain rate is less than 20% of the mechanical damping rate, and hence we do not expect or observe notable contributions to the noise from optomechanics.

3. In general, its not appropriate to count gain towards efficiency if one has quantum applications in mind. Gain adds noise, which results in mixed states. Gain should be divided out of reported efficiencies.

We agree that this important detail was missing from the original manuscript. We have added a more full description of the quantum efficiency in which the gain is subtracted out and the decoherence due to optomechanical gain is included. Figure S4b has been modified to include both the classical and quantum efficiencies, and the following discussion was added to the supplementary material:

In the quantum regime, calculations of the efficiency are more complicated. The parametric driving, which can allow for efficient classical transport, also introduces extra noise. Furthermore, quantum states with small phonon occupation have a large thermalization rate due to the high thermal occupation of the bath, even when the resonator is cooled down to mK temperatures. In Figure S4b, we compare the classical and quantum efficiencies. At small detuning the quantum efficiency is limited by parametric driving and at large detuning by thermalization.

4. Similarly, in future projections, the 99% efficiency should have gain divided out. Perhaps in the quantum regime, gain may be desirable as it implies entanglement with the other mechanical mode, but this is a separate issue from state transfer and should be discussed separately.

We have adjusted our future projections in the main text to include the modified quantum efficiency:

In the quantum regime, the negative component of Γ introduces extra decoherence, so the quantum state transfer is more limited (56% efficiency). However, the effects of coherent swapping should still be visible.

5. In Fig. 3b, the Resonator 1 curve first peaks at ~50 pm. For identical oscillators, one expects it to peak around ~100 pm. Is the balance explained by a difference in mechanical parameters? Naively, Resonator 1 has the lower frequency, so I expect its first peak to be higher, not lower.

The balance is described by different mechanical parameters. In particular, the lower frequency resonator has a much larger effective mass, which means it moves less for the same mean phonon occupation. As the reviewer stated, the lower mechanical frequency would tend to make the motion larger, but this is not enough to outweigh the effect of the effective mass.

Reviewer #3 (Remarks to the Author):

The authors report the realization of coupling non-degenerate mechanical oscillators using a shared optical cavity mode. In addition to the coupling of two non-degenerate fundamental modes they demonstrate coupling involving mechanical modes with higher frequency. These results are of great importance to the optomechanics community for the first demonstration of coherently mediated coupling, but the impact extends beyond optomechanics, as set-ups modelled after these results may be useful in future high-precision measurements and in experiments investigating collapse models of quantum mechanics. I judge the impact of their results to be sufficient to warrant publication in Nature communications, if the authors clarify some important details.

We thank the reviewer for the careful analysis and the positive recommendation. We agree that these results could be used in many diverse experiments, and that this method will be particularly useful for future experiments in optomechanics.

Most importantly, the authors claim their transfer to be coherent. However, the data they show are measurements of the mechanical mode's energies. Moreover, neither the main body, the methods or supplemental section contain the procedure through which these energies are obtained. This is unsatisfactory and needs to be addressed.

We have clarified the method we use to measure the amplitude of motion in the supplemental materials:

One laser is locked to the cavity resonance with the Pound-Drever-Hall technique using an avalanche photo diode as a detector and the error signal is sent to two lock-in amplifiers, each of which monitors one mechanical frequency and extracts the amplitude of motion of the corresponding resonator. Before the swapping experiment shown in Figure 3 is performed, we calibrate the mechanical motion of the devices by measuring the thermal motion for approximately one minute.

Related to this question, the authors mention in the main body that the thermal environment of the system lies in the mK range. How is it cooled? Did the authors try to further cool the mechanical degrees of freedom using sideband cooling? Their system is in a regime where this would be an efficient technique.

We understand how the manuscript was a little vague, and could have led to the impression that the experiment was performed at mK temperatures. However, the system is not cooled in the present experiment. We have clarified the experimental conditions by stating that the system is at room temperature in the results section on page 4 and the discussion section on page 6. We

do, however, propose cooling the system to the mK regime in the future. This can be done using a combination of dilution refrigeration and optical sideband cooling as the reviewer suggests.

The thermal noise floor in Fig. 3b is not discernible, it would be helpful to have an inset around the time of one completed transfer. In the same figure, are the lines from the theory model presented? what causes the initial offset?

We have added an inset to Fig. 3b, as suggested. The lines from the theory model are shown in supplementary materials IV, equations 13 and 14. We are a little uncertain what the “initial offset” refers to. If it is the amplitude of motion of resonator 2, this is determined by the dielectric driving of the resonator. If this refers to the small deviation from the theory in the first 200 ms, it is well within the bounds of the thermal fluctuations. In Fig. 3a the measured amplitude of motion of the resonators demonstrates the typical expected thermal fluctuations on top of the average.

In fig. 4 data is compared to theory curves resulting from a "2 parameter fit". Which parameters were used as fitting parameters? All parameters in the presented model can be measured independently and the authors suggest that they in fact did measure them independently.

The two parameters are the input power (P_{in}) and the mean bare mechanical dissipation rate ($(\gamma_1/2 + \gamma_2/2)$). The input power is quite difficult to measure directly, and the ratio between the optical power we measure and the input power can vary from day to day. However, we can confirm that the input powers we fit are reasonable by comparing to our data from the optical cooling experiments shown in figure S1b. The readout laser (locked with PDH) can slightly increase or decrease the mechanical linewidth, depending on the exact settings of the lock. We observed that there is some variation from sweep to sweep.

We add to the caption of Figure 4:

The ratio between the measured optical power and the input power P_{in} and the mean bare mechanical dissipation rate $(\gamma_1 + \gamma_2)/2$ are the fitting parameters (see Supplementary III.)

And to the supplementary materials:

We can't measure the optical input power directly, so we split off some power before the cavity to measure. The readout laser can optomechanically decrease or increase the bare mechanical linewidths of the resonators a small amount depending on the lock settings. We therefore fit the mean bare mechanical linewidth and the ratio between the measured optical power and the input power for every sweep shown in Figure 4.

If these comments will be taken care of, I will be happy to support publication of the author's results.

REVIEWERS' COMMENTS:

Reviewer #2 (Remarks to the Author):

The authors have now adequately reported and discussed the effects of gain in their experiment, including in projections for future work.

I find that the manuscript is now suitable for publication.

Reviewer #3 (Remarks to the Author):

The authors have considered my and the other referee's comments, which resulted in an improved manuscript.

I am satisfied with the reviewed version and recommend publication.